Sevoflurane preconditioning attenuates hypoxia/reoxygenation injury of H9c2 cardiomyocytes by activation of the HIF-1/PDK-1 pathway

Hou Tianliang 1
Ma Haiping 1
Wang Haixia 2
Chen Chunling 1
Ye Jianrong 616227972@qq.com 1
Ahmed Ahmed Mohamed 3
Zheng Hong xj_zhenghong@sina.com 1
1 Department of Anesthesiology, The First Affiliated Hospital of Xinjiang Medical University , Urumqi , Xinjiang , China
2 Department of Mastology, Xinjiang Maternal and Child Health Hospital , Urumqi , Xinjiang , China
3 Department of Intensive Care Unit (ICU), Yardimeli Hospital , Mogadishu , Somalia
Strohecker Anne
Electronic publication date: 2020 Dec 21
Publication date: 2020
Volume: 8
Electronic Location ID: e10603
Received 2020 Sep 3; Accepted 2020 Nov 28
Copyright: ©2020 Hou et al.
Copyright year: 2020
Copyright holder: Hou et al.
License: This is an open access article distributed under the terms of the Creative Commons Attribution License, which permits unrestricted use, distribution, reproduction and adaptation in any medium and for any purpose provided that it is properly attributed. For attribution, the original author(s), title, publication source (PeerJ) and either DOI or URL of the article must be cited.
License URL: https://creativecommons.org/licenses/by/4.0/

Keywords: Sevoflurane preconditioning, Myocardial protective effect, Hypoxia-inducible factor-1, Glycolysis, Hypoxia/reoxygenation injury, Pyruvate dehydrogenase kinase-1

Funding: The National Natural Science Foundation of China U1603129 81660522 This work was supported by the National Natural Science Foundation of China (No. U1603129) and the National Natural Science Foundation of China (No. 81660522). The funders had no role in study design, data collection and analysis, decision to publish, or preparation of the manuscript.

==============================
Background

Sevoflurane preconditioning (SPC) can provide myocardial protective effects similar to ischemic preconditioning (IPC). However, the underlying molecular mechanism of SPC remains unclear. Studies confirm that hypoxia-inducible factor-1 (HIF-1) can transform cells from aerobic oxidation to anaerobic glycolysis by activating the switch protein pyruvate dehydrogenase kinase-1 (PDK-1), thus providing energy for the normal life activities of cells under hypoxic conditions. The purpose of this study was to investigate whether the cardioprotective effects of SPC are associated with activation of the HIF-1a/PDK-1 signal pathway.

Methods

The H9c2 cardiomyocytes hypoxia/reoxygenation model was established and treated with 2.4% sevoflurane at the end of equilibration. Lactate dehydrogenase (LDH) level, cell viability, cell apoptosis, mitochondrial membrane potential, key enzymes of glycolysis, ATP concentration of glycolysis were assessed after the intervention. Apoptosis related protein(Bcl-2, Bax), HIF-1a protein, and PDK-1 protein were assessed by western blot.

Results

Compared with the H/R group, SPC significantly increased the expression of HIF-1a, PDK-1, and Bcl-2 and reduced the protein expression of Bax, which markedly decreased the apoptosis ratio and Lactate dehydrogenase (LDH) level, increasing the cell viability, content of key enzymes of glycolysis, ATP concentration of glycolysis and stabilizing the mitochondrial membrane potential. However, the cardioprotective effects of SPC were disappeared by treatment with a HIF-1a selective inhibitor.

Conclusion

This study demonstrates that the cardioprotective effects of SPC are associated with the activation of the HIF-1a/PDK-1 signaling pathway. The mechanism may be related to increasing the content of key enzymes and ATP of glycolysis in the early stage of hypoxia.

Introduction

Despite substantial progress in prevention and treatment, ischemic heart disease (IHD) remains the main cause of morbidity and mortality worldwide (Devereaux et al., 2017). At present reperfusion therapy is the most effective treatment for IHD. However, it is also inevitable that ischemia/reperfusion (I/R) injury will occur to cause greater injury for tissues (Heusch, 2013). Accumulated evidence has indicated that preconditioning with sevoflurane could attenuate the I/R injury both in vitro and in vivo (Guerrero-Orriach et al., 2017; Kunst & Klein, 2015; Obal et al., 2005), and there are multiple mechanisms involved in the cardioprotective effect of sevoflurane preconditioning (SPC), such as activation of KATP channel, to improve ion transport in the myocardial cell membrane, mitochondrial membrane and relieve Ca2+ overload (Zhang et al., 2012), activation of Adenosine A1 receptor as to reduce the production of reactive oxygen species (Lucchinetti et al., 2018), downregulation of apoptosis (Wei et al., 2017). However, the potential molecular mechanisms of SPC in myocardial protection still have not been fully elucidated.

Hypoxia-inducible factor-1 (HIF-1) is a protein that regulates the expression of an arrangement of hypoxia-sensitive proteins beneath hypoxic conditions to preserve the survival of tissues (Semenza, 2011). Considers have affirmed that HIF-1 is central to cardio-protection against I/R injury (Eckle et al., 2008). Pyruvate dehydrogenase kinase (PDK) is a key protein downstream of HIF-1 regulating energy metabolism. Now four isozymes of PDK-1, PDK-2, PDK-3, and PDK-4 have been identified in PDK, they are mainly located in the mitochondrial matrix, in which PDK-1 is mainly expressed in the heart (Sugden & Holness, 2003). When cells are in a state of hypoxia, HIF-1 regulates the expression of downstream target protein PDK-1, which changes the energy metabolism of cardiomyocytes from the tricarboxylic acid cycle to glycolysis (Holness & Sugden, 2003). Although glycolysis is an inefficient way of energy production compared with mitochondrial aerobic oxidation, it plays an important role in sustaining basic life activities under the conditions of ischemia/hypoxia. Thus, we investigated whether the myocardial protective effect of SPC was associated with the regulation of glycolysis activity by activating the HIF-1a/PDK-1 signaling pathway in H9c2 cardiomyocytes.

In the present study, H9c2 cardiomyocytes were used to establish a hypoxia/reoxygenation (H/R) induced myocardial cell injury model. To explore the molecular mechanism of SPC against H/R injury of cardiomyocytes by studying the activities of key enzymes and energy metabolism of glycolysis.

Material and Methods

Reagent and antibodies

TRYPSIN 0.25% (1X) solution with EDTA, penicillin-streptomycin solution and DMEM/low glucose medium, phosphate-buffered saline (PBS) were purchased from Hyclone (Cat#SH30042.01, SV30010, SH30021.01, SH30256.01; Hyclone, US). Fetal Bovine Serum (Cat#10091148, GIBCO, New York, US). Dimethyl sulfoxide, Tris base, Glycine were purchased from Sigma (Cat# V900090, V900483, V900144; Sigma, US).

Annexin V-PE apoptosis detection kit was purchased from Becton, Dickinson, and Company (Cat#559763, US). Mitochondrial membrane potential detection kit (JC-1), 10 ◊TBST buffer, non-fat milk powder, sodium dodecyl sulfate, SDS-page gel preparation kit, 4 ◊protein sample buffer (containing β-mercaptoethanol) were purchased from Solarbio (Cat#M8650, T1081, S8010-100, P1200-2, P1016; Solarbio, Beijing, China). ATP detection kit and BCA protein quantitative kit were purchased from Beyotime (Cat#s0026, P0012, Beyotime, Shanghai, China). The lactate dehydrogenase (LDH) kit was purchased from Nanjing Jiancheng Biotech Co. Ltd (Cat# A020-2).

Rabbit-anti-HIF-1 alpha monoclonal antibody [EPR16897] (ab179483), Rabbit-anti-GAPDH antibody [EPR16891] (ab181602-100), Rabbit-anti-Bcl-2 monoclonal antibody (ab62557), Rabbit-anti-Bax monoclonal antibody (ab32503), horseradish peroxidase (HRP)-conjugated goat anti-rabbit immunoglobulin G (IgG) (ab205718), 10 ◊RIPA lysate (ab156034), Protease Inhibitor Cocktail, (ab201111), ECL Substrate Kit (High Sensitivity) (ab133406) were purchased from Abcam Cambridge, MA, US. Rainbow protein pre-staining marker, 10 to 180 kDa (26616; Thermo-Fermentas, MA, US). Rabbit-anti-PDK-1 (Cat# BF0312; Affinity Biosciences, Changzhou, China). Lificiguat (YC-1) was purchased from Selleck (Cat#S7958,Shanghai, China).

Cell culture and processing

The H9c2 rat embryonic cardiomyocyte cell line was obtained from Procell Life Science & Technology Co., Ltd., China. The cell culture conditions consisted of DMEM (low sugar, 5 mM) medium+10% (v/v) FBS (Gibco, USA) + 1% (v/v) Penicillin/Streptomycin solution (Gibco, USA) at 37 °C, 5% CO2, and saturated humidity. The H9c2 cells were passaged at the ratio of 1:4 when the cell confluence reached 90% (by visual estimate) in 55 cm2 culture plate. Cells were cultured for 48 h at 37 °C in a 5% CO2 incubator. The supernatant was discarded when the cells grew to 90% confluency, After gently washing the adherent cells with PBS twice, the serum-free DMEM (low sugar, 5 mM) medium was added. Treatment with Lificiguat (YC-1, 10 µM) was added before sevoflurane preconditioning. Plates with H9c2 cardiomyocytes were then placed in a sealed chamber (Modular Incubator Chamber, MIC1; Billups-Rothenberg, Inc., Del Mar, CA, USA; http://www.brincubator.com/) filled with 95% N2 and 5% CO2 to achieve an oxygen-deficient environment. Ventilation at 5 L/min for 15 min was used to achieve a 1% lower oxygen concentration in the chamber (The oxygen indicator card was used to reflect the oxygen concentration. Its color will change from blue to red when the oxygen concentration is less than 0.1% in the sealed chamber). Cells were incubated at 37 °C for 3 h, and the PBS was removed and replaced with fresh medium containing 10% FBS. Longer 3 h incubation in 95% air and 5% CO2 at 37 °C was performed as reoxygenation. The plates of the CON group were kept in normoxic conditions for the corresponding times.

Experimental protocol

The H9c2 cardiomyocytes were randomly divided into four groups (Fig. 1).

Figure 1 A schematic representation of the experiment protocol.

H9c2 cardiomyocytes were randomly divided into control (CON), hypoxia/reoxygenation (H/R), sevoflurane preconditioning (SPC) and sevoflurane preconditioning+HIF-1a inhibitor (S+Y) groups.

(1) Control (CON) group: H9c2 cells were continuously cultured in DMEM low-glucose medium (5 mM) containing 10% FBS without any interventions but with inhalation of pure oxygen (100% oxygen) for 15 min at the same time interval when the SPC group was exposed to sevoflurane.

(2) Hypoxia/reoxygenation (H/R) group: H9c2 cells were cultured in low glucose concentration medium for 48 h, removed, washed with PBS two times and inhaled pure oxygen (100% oxygen) for 15 min and placed in an airtight container with 95% N2 and 5% CO2 for 3 h, then reoxygenation (95% air and 5% CO2) with the addition of fresh low glucose DMEM with 10% FBS at 37 °C, for a total 3 h of reoxygenation.

(3) Sevoflurane preconditioning group (SPC): H9c2 cells were exposed to 2.4% sevoflurane for 15 min before hypoxia after incubation in the medium for 48 h. the later steps are all same to the H/R group.

(4) Hypoxia-inducible factor-1a inhibitor YC-1 group (S+Y): Lificiguat (YC-1) was added before sevoflurane preconditioning, after treatment with sevoflurane preconditioning, the later steps are all same to the SPC group.

Measurement of lactate dehydrogenase(LDH)

Data were collected as described previously (Yu et al., 2016). Cell supernatant (0.1 ml) was collected from each group’s plate 1 min after reoxygenation. The absorbance values were measured at 450 nm, and then put the absorbance values into the formula (LDH (U/L)=[(Determination OD-Control OD)(Standard OD-Blank OD)] ×0.2 µmol/ml ×1,000), LDH activity was expressed as international units per liter (IU/L).

Measurement of cell viability

Data were collected as described previously (Yang et al., 2019). Each experiment was repeated three times, and each group was repeated twice.

Flow cytometry assay

The percentage of cells apoptosis was measured using the PE Annexin-V Apoptosis Detection Kit I (BD Biosciences Pharmingen, US). In brief, cells were collected and stained with annexin-V and 7-AAD as per the manufacturer’s instrument. The apoptotic cells were identified by flow cytometry (Beckman Coulter, US).

Detection of key enzymes in glycolysis

The content of the key enzymes of glycolysis was detected by the Elisa High-end detection kits (Jianglai Biotech Co. Ltd, China). The key enzymes of glycolysis, including that of Hexokinase (HK), 6-phosphofructokinase-1 (PFK-1), Pyruvate kinase (PK), were measured. Wash the cells twice with pre-cooled PBS. RIPA lysis buffer was used to lyse the cells on ice for 30 min, then the cell lysate was collected by EP tube. Centrifugation: 16,000 rpm, 4 °C, 10 min, collecting the supernatant, and waiting for the test. The standard curve was made on enzyme coated plate with 50 µl different concentrations of a standard substance. 40 µl of sample diluent was added to the enzyme coated plate, and then 10 µl of the sample to be tested was added. 100 µl of enzyme-labeled reagent was added into each well. The mixture was incubated at 37 °C for 60 min, then washed 5 times and left to dry. Chromogenic agent solutions A and B (50 µl each) were added and incubated at room temperature for 15 min in the dark. After adding 50 µl of the termination solution, the OD values of each pore were measured by an Enzyme labeling instrument at 450 nm.

Detection of glycolysis energy

The level of intracellular ATP was detected using the ATP Bioluminescence Assay Kit (Beyotime, Shanghai, China). The cells lysate was added, fully lysed, and centrifuged at 15,000 rpm for 10 min at 4 °C. The supernatant was collected into an EP tube, waiting for the test. Take 200 µl ATP detection reagent and add 1,800 µl ATP detection reagent diluent to prepare a 2 ml ATP detection working solution. The ATP standard solution was diluted into an appropriate concentration gradient with ATP detection lysate and made a standard curve using concentrations of 0.01, 0.03, 0.1, 0.3, 1, 3 and 10 µM. After adding 100 µl ATP detection working solution and 20 µl sample into the detection well, and then measured with multifunctional enzyme marker after two seconds interval.

Detection of mitochondrial membrane potential

JC-1 (Millipore, USA) is used to assess the redistribution of mitochondrial membrane potential (ΔΨm). According to the manufacturer‘s instruction, the H9c2 cells were incubated with 10 nmol/L JC-1 staining solution at 37 °C free light for 10 min. The images were captured using a fluorescence inverted microscope (LEICA-DMI4000B, Germany). Thirty randomly chosen cardiomyocytes per treatment group were analyzed (n = 3 independent experiments with 10 incubated cardiomyocytes per experiment). Red and green fluorescence intensities were analyzed respectively using Image J software as described (Sun et al., 2019), and the ratio of the red/green fluorescence was proportional to the ΔΨm.

Western blot analysis

The HIF-1a, PDK-1, Bax, Bcl-2, and glyceraldehyde 3-phosphate dehydrogenase (GAPDH) proteins were detected with the antibodies rabbit monoclonal anti-HIF-1a, rabbit monoclonal anti-GAPDH, rabbit monoclonal anti-Bax, rabbit monoclonal anti-Bcl-2 (Abcam, US), rabbit monoclonal anti-PDK-1 (Affinity, USA), Proteins were extracted with 100 µl Lysis Solution (Radioimmunoprecipitation assay buffer/Phenylmethanesulfonyl fluoride (PMSF)=100:1) for each plate. The protein concentrations were measured using a BCA kit (Solarbio, Beijing, China) and all the concentrations were adjusted to the lowest protein concentration with the RIPA lysate. The supernatant was mixed with 5 ◊ loading buffer and heated for 5 min at 100 °C, and then 30 micrograms of the sample were subjected to electrophoresis using an SDS-PAGE gel system, transferred to Polyvinylidene fluoride (PVDF) Western blotting membranes (Roche, Germany) and blocked in 5% non-fat milk at 37 °C for 2 h. Diluted primary antibodies to HIF-1a [EPR16897] (1:1000, Abcam, ab179483), Bax (1:500, Abcam, ab32503), GAPDH (1:1000, Abcam, ab181602-100), PDK-1 (1:1000, Affinity Biosciences Cat#BF0312) and Bcl-2 (1:500, Abcam, ab62557) were added, and the membrane was incubated overnight (4 °C). The membrane was washed with TBST solution for 3 times and incubated with HRP-conjugated secondary antibody (1:10,000, Abcam, ab205718) for 2 h at room temperature. High Sensitivity ECL Substrate Kit (Abcam, ab133406) was used for visualization and imaging. Signals were detected and quantified with Image Lab 4.0 software (Bio-Rad Laboratories, US).

Statistical analysis

SPSS 20.0 statistical software was used for the data analysis. All values were expressed as the mean ± standard deviation. The within-group comparisons were performed using the analysis of variance of repeated measurement design. Pairwise comparison in multiple groups was conducted with SNK method. P < 0.05 was considered significant. GraphPad Prism 5.0 was used to prepare graphs.

Results

SPC reduced LDH level, cell death and expression of protein Bax, increased cell viability and expression of protein Bcl-2 following simulated Hypoxia/Reoxygenation (H/R)

To observe the effect of SPC on H/R induced cell injury, the H9c2 cells were subjected to LDH level, cell viability, flow cytometry, and western blot analysis. In this study, The cell viability of the H/R group was significantly lower than the CON group (P < 0.05), while the cell viability of the SPC group was increased compared to the H/R group (P < 0.05, Fig. 2A). The LDH assay showed that compared to the H/R group, the LDH activity of the SPC group was decreased (P < 0.05), while the LDH activity did not significantly differ between the H/R and S+Y groups (P > 0.05, Fig. 2B). Compared to the CON group, the cells apoptotic rate of H/R and S+Y groups were significantly increased (P < 0.05), while the cells apoptotic rate of the SPC group was decreased compared to the H/R group (P < 0.05, Fig. 2C). Flow cytometry to measure apoptosis distribution graphs were shown in Figs. 2D–2G. The expression levels of apoptosis-related proteins were measured by using western blot analysis. Western blot images of Bcl-2 protein and Bax protein were shown in Figs. 2H–2I. Compared to the CON group, the ratio of Bcl-2/GAPDH of H/R and S+Y groups did not have a significant difference (P > 0.05), however, the ratio of Bcl-2/GAPDH of SPC group was significantly increase (P < 0.05, Fig. 2J). Compared to the CON group, the ratio of Bax/GAPDH of H/R and S+Y groups were significantly increased, but the ratio of Bax/GAPDH of SPC was decreased (P < 0.05, Fig. 2K).

Figure 2 SPC alleviated H9c2 cardiomyocytes hypoxia/reoxygenation injury.

(A) Cell viability: Compared to the H/R group, the SPC group exhibited improved cell viability (n = 6). (B) LDH activity: The LDH activity was reduced in the SPC group compared to the H/R group (n = 6). (C) Apoptosis rate: The apoptosis rate of the SPC group was lower than that of the H/R group. While the protective effects of SPC were inhibited after administration of YC-1 (n = 6). (D–G) Flow cytometry to measure apoptosis distribution graph. (H) Western blot image of Bcl-2 protein. (I) Western blot image of Bax protein. (J) Bcl-2 protein expression. (K) Bax protein expression. Protein content was normalized to GAPDH (n = 3/group). Data represent mean ± SD (*P < 0.05 vs C group, #P < 0.05 vs H/R group, &P < 0.05 vs SPC group).

Sevoflurane preconditioning increases the activity of key enzymes and ATP production in glycolysis

The HK, PFK-1, and PK activities at the end of equilibration did not significantly differ among groups (P > 0.05). The HK, PFK-1, and PK activities at the end of reoxygenation were significantly increased in all groups compared to the activities at the end of equilibration (P < 0.05). The comparison of all groups at the end of reoxygenation showed that the HK, PFK-1, and PK activities in the SPC group were significantly increased compared to those in the H/R group (P < 0.05, Figs. 3A, 3B and 3C).

Figure 3 SPC improve glycolysis energy metabolism and key enzymes activity.

The end of equilibration (T1) and the end of reoxygenation (T2). (A) HK concentration. (B) PFK-1 concentration. (C) PK concentration. (D) ATP concentration. Data are presented as the mean ± SD (n = 6). *P < 0.05 vs T1; T2 time point: ˆP<0.05 vs CON, #P < 0.05 vs H/R, &P < 0.05 vs SPC.

The concentration of ATP at the end of equilibration did not significantly differ among groups (P > 0.05). The ATP assay showed that compared with the Con group, the ATP concentration of H/R, SPC and S+Y groups were significantly decreased (P < 0.05). Compared to the H/R group, the ATP concentration of the SPC group was increased, while the S+Y group was decreased (P < 0.05, Fig. 3D).

Protective effects of SPC on H/R injury of H9c2 cells

ΔΨm, a sign of early-stage apoptosis (Marchetti et al., 1996) was evaluated in H9c2 using JC-1 staining. Representative images of JC-1 in H9c2 cells were shown in Fig. 4A. The result showed that the ratio of red to green fluorescence intensity in the SPC group was higher than the H/R group (P < 0.05). However, After the application of hypoxia-inducible factor-1 α inhibitor YC-1, there was no significant difference in the ratio of red to green fluorescence intensity between H/R group and S+Y group (P < 0.05, Fig. 4B).

Figure 4 SPC stabilize the mitochondrial membrane potential (ΔΨ m).

(A) Representative images of JC-1 were capture using fluorescence inverted microscope in H9c2 cells. (B) The results showed that SPC increased the ratio of red to green fluorescence intensity. Data represent mean ± SD (n = 3/group). *P < 0.05 vs Con, #P < 0.05 vs H/R, &P < 0.05 vs SPC.

SPC increases the levels of HIF-1a and PDK-1 in the simulation of H/R injury

At the end of reoxygenation, the expression of HIF-1a of SPC and H/R groups were all increased compared to the Con group (P < 0.05). Western blot images of HIF-1a protein and PDK-1 protein were shown in Fig.A-B. Compared to the H/R group, the expression of HIF-1a of the SPC group was increased, while the S+Y group was decreased (P < 0.05, Fig. 5C). Compared to the CON group, the expression of PDK-1 of SPC and H/R groups were all increased (P < 0.05), while the S+Y group did not have statistical significance (P > 0.05). Compared to the SPC group, the expression of PDK-1 of the S+Y group was decreased (P < 0.05) when the inhibitor YC-1 was added into the culture medium (Fig. 5D).

Discussion

The results of our study confirmed that sevoflurane preconditioning (SPC) can effectively attenuate the Hypoxia/Reoxygenation (H/R) injury in H9c2 cardiomyocytes. This finding is in line with the proven SPC cardiac protective effects reported in a related study (Wenlan et al., 2018). It is suggested that the cardioprotective effect of SPC may be related to the activation of the HIF-1a/PDk-1 pathway.

Sevoflurane is widely used in all types of clinical cardiac surgery as a representative inhalation anesthetic drug and is considered by the majority of scholars to be a landmark inhalation anesthesia drug. Some studies have found that the protective effect of sevoflurane on ischemic myocardium, such as sevoflurane preconditioning (SPC) or sevoflurane postconditioning (SPostC), is closely related to its mode of administration (De Hert et al., 2004; Qian et al., 2018; Wu et al., 2017). Ischemic preconditioning (IPC) is the most effective myocardial protection strategy found so far (Cai et al., 2003), and SPC can produce myocardial protective effects similar to IPC (Qian et al., 2018) and has stronger clinical maneuverability. In the study, H9c2 cardiomyocytes were used as the research object to simulate the H/R model to explore the molecular mechanism of myocardial protection of SPC, to provide a theoretical basis for scientific and clinic research.

Figure 5 SPC upregulate the protein expression of HIF-1a and PDK-1.

(A) Western blot image of HIF-1a protein. (B) Western blot image of PDK-1 protein. (C) HIF-1a protein expression. (D) PDK-1 protein expression. Protein content was normalized to GAPDH. Data represent mean ± SD (n = 3/group). *P < 0.05 vs Con, #P < 0.05 vs H/R, &P < 0.05 vs SPC.

A member of the hypoxia-inducible factor family (HIFs) is HIF-1. Due to the inhibition of the activity of prolyl-4-hydroxylase domain-containing enzymes (BHD) under anoxia situations, the ubiquitination of HIF-1 is reduced. In the nucleus, it binds to the target gene promoter’s hypoxia response element and regulates the transcription of a series of downstream target genes which contribute to cell survival under hypoxic conditions (Prabhakar & Semenza, 2015). PDK-1 is a switch protein that regulates energy metabolism downstream of HIF-1 (Semenza, 2011). Glucose will form pyruvate under aerobic conditions and then enter into the tricarboxylic acid cycle to be oxidized to carbon dioxide and water, while in the anaerobic condition, pyruvate undergoes glycolysis under the action of PDK-1 to produce lactic acid (Akram, 2013). In this study, SPC and HIF-1a inhibitor YC-1 were used as intervention methods for grouping study. Western Blot protein detection technique was used to quantify the contents of target protein HIF-1a and PDK-1. The results showed that the contents of HIF-1a and PDK-1 were significantly increased in the SPC group, while the contents were significantly decreased after using the inhibitor of HIF-1a. It is suggested that the myocardial protective effect of SPC may be related to activation of the HIF-1a/PDK-1 signal pathway.

ATP is the direct energy for all kinds of life activities. The myocardium has a high energy demand, but there is basically no energy reserve (Lopaschuk et al., 2010). Therefore, the heart must continuously produce a large amount of ATP to maintain muscle contraction and ion homeostasis. Most of the ATP production (about 90–95%) comes from oxidative phosphorylation of mitochondria and the rest from glycolysis. Mitochondria can use a variety of energy substrates to produce ATP, including fatty acids, carbohydrates, ketones and amino acids, among which fatty acid oxidation is the main source of ATP in the heart (Itoi & Lopaschuk, 1993). What’s more, normal myocardium has a lot of “metabolic flexibility” (Kolwicz Jr, Purohit & Tian, 2013; Ritterhoff & Tian, 2017), allowing it to switch back and forth between fatty acid and carbohydrate oxidation, depending on the load of the heart, the energy substrate provided to the heart, and the hormonal and nutritional status (Taegtmeyer et al., 2004). The contribution for myocardial ATP production by glycolysis was increased under the condition of myocardial ischemia/hypoxia (Akram, 2013). Glycolysis is a dynamic process, which provides ATP for the body in the early stage of hypoxia. However, with the extension of time, glycolysis will produce excessive acid metabolites, which will lead to acidosis and aggravate the apoptosis and damage of the body (Dang, Huang & Zhou, 2005). The study confirmed that the glycolytic pathway content of key rate-limiting enzymes (HK, PFK-1, and PK) in cardiomyocytes may indirectly reflect the state of energy metabolism of cardiomyocytes (Volker, Reinitz & Knull, 1995). And after hypoxia injury, the levels of HK, PFK-1, and PK increased and the glycolytic activity of cardiomyocytes was significantly increased in early hypoxia (Guo et al., 2017; Teng et al., 2010) and cardiomyocyte glycolysis activity reached its peak at 1–3 h of hypoxia, Since then, and has been decreasing (Dang, Huang & Zhou, 2005). In our experiment, the time limit of cell hypoxia damage was set as 3 h and the results showed that activity of the key enzymes of glycolysis of H/R, SPC and Y+S groups were significantly increased compared to CON group, and the activity of the key glycolytic enzymes was the highest in the SPC group. The ATP concentration of SPC was also higher than the H/R group. There are evidences that HIF-1a is also the main regulator of glycolytic enzyme expression and the key transcription factor that drives glycolysis under anoxic conditions (Semenza, 2010; Xie et al., 2015). These evidences led us to hypothesize that SPC could optimize the glycolysis in early hypoxia by activating the HIF-1a/PDK-1 signal pathway.

Mitochondria are not only the main place of energy metabolism in eukaryotes but also the main target of ischemic/hypoxic injury (Halestrap & Richardson, 2015). The level of mitochondrial membrane potential is an index of early cardiomyocyte apoptosis (Marchetti et al., 1996). In this study, JC-1 was used for cell staining, and the levels of mitochondrial membrane potential in different experimental groups were compared and the apoptosis-related proteins Bcl-2 and Bax were detected by WB. The results showed that the levels of anti-apoptotic protein Bcl-2 and mitochondrial membrane potential in the SPC group were higher than those in H/R and Y+S groups, indicating that SPC could stabilize cardiomyocyte mitochondrial membrane potential and reduce the rate of apoptosis induced by H/R injury in H9c2 cardiomyocytes.

Limitations

Several limitations of this study should be noted. Firstly, we only tested the hypothesis of the cardioprotective effect of SPC at the cardiomyocyte level, and it should be further verified at the animal level in the future. Secondly, the concentration of SPC used in this study was 2.4%, and the effect of other concentrations on the results should also be investigated. Thirdly, we only observed the potential HIF-1a/PDK-1 signaling pathways mechanisms in the myocardial protective function of SPC, further investigation is needed to identify essential components in these complex signal transduction cascades that mediate SPC.

Conclusions

In summary, this study demonstrated that the cardioprotective effect of SPC are associated with the activation of the HIF-1a/PDk-1 pathway. The mechanism may be related to an increase in the expression of HIF-1a, PDK-1 and Bcl-2 after SPC, which leads to increasing the contents of key enzymes and ATP concentration of glycolysis, thereby stabling mitochondrial membrane potential and reducing apoptosis.

Supplemental Information

Supplemental Information 1 Original strip.

1.CON, 2.HR, 3.SPC, 4.S+Y, 5.CON, 6.HR, 7.SPC, 8.Y+S.

Click here for additional data file.

Supplemental Information 2 The within group comparisons were performed using the analysis of variance of repeated measurement design. The comparison between groups was performed using one-way analysis of variance

Click here for additional data file.

Additional Information and Declarations

Competing Interests

Author Contributions

Data Availability

The authors declare there are no competing interests.

Tianliang Hou conceived and designed the experiments, performed the experiments, analyzed the data, prepared figures and/or tables, authored or reviewed drafts of the paper, and approved the final draft.

Haiping Ma conceived and designed the experiments, performed the experiments, authored or reviewed drafts of the paper, and approved the final draft.

Haixia Wang performed the experiments, prepared figures and/or tables, and approved the final draft.

Chunling Chen and Hong Zheng analyzed the data, authored or reviewed drafts of the paper, fund support, and approved the final draft.

Jianrong Ye and Ahmed Mohamed Ahmed analyzed the data, authored or reviewed drafts of the paper, and approved the final draft.

The following information was supplied regarding data availability:

The raw measurements and blots are available in the Supplemental Files.

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
