# Peer review of "Sevoflurane preconditioning attenuates hypoxia/reoxygenation injury of H9c2 cardiomyocytes by activation of the HIF-1/PDK-1 pathway"

_PeerJ, doi:10.7717/peerj.10603_

## Round 0.1 · original submission · Minor Revisions

Your manuscript has been reviewed by two expert Reviewers and an Academic Editor. Our comments are presented below.

The Reviewers indicate that experiments were generally sound, well controlled, and hypothesis-driven. However, both noted that the data presented do not compellingly support upregulation of glycolysis as the mechanism of action of Sevoflurane-induced reduction H/R injury, but rather are supportive of a role of HIF modulation. Please revise to address this critique. Additionally, the use of an editorial service to improve grammar and readability of the work is recommended. Peer J does not provide copy editing assistance at this time.

Several other minor corrections were requested (outlined below). Please carefully consider these comments as you revise the work. I look forward to receiving your revised manuscript.

·

Basic reporting

This manuscript needs to be edited by a reader with an eye towards grammar and English language usage - the science is sound, but it is not well conveyed in English. That said, this is a straightforward manuscript, with a clear hypothesis supported by experiments described.

Experimental design

The experiments are well designed. The controls are appropriate, and the use of multiple endpoints to evaluate cell survival strengthens the conclusions. H9c2s are a limited but appropriate model. The methods are very thorough.

Validity of the findings

The data is provided, and findings are valid, with the caveat that the interpretation of the findings is at times a bit too broad. In particular, H9c2s, although derived from rat cardiomyocytes, are metabolically distinct from primary cardiomyocytes. While this is noted in the limitations, at times in the rest of the manuscript this distinction is blurred (for example, the title should clarify that H9c2 cells are used). Also, the authors should be careful to note the data provided do not demonstrate that SEV-induced upregulation of glycolysis via HIF1a is responsible for the difference in survival rate. HIF blockade may protect cells, and a side effect may be that it upregulates glycolysis.

Additional comments

1. The introduction is overly descriptive and may be shortened. For example, it is not relevant to mention the Nobel Prize in lines 42-44, or the origin of glycolysis studies in yeast in lines 50-51.
2. How is final O2 (listed as <1%, I believe) measured, and controlled during hypoxia?
3. Fig 1 – please change the length of the bars to better reflect the duration (i.e. 15 min should be 1/12th the length of the 3 hr incubations. The 48 hr equilibration should be a split bar etc.) Also, please label T1 and T2 here for reference to Fig 3
4. Please clarify that the Fig 2 flow diagrams are annexin – labeled
5. Fig 2E: the Bax represenative blot looks overexposed, as well as the blots in Figure 5B. Please choose a less-exposed blot for these images.
6. Figure 4: please state what red and green represent.
7. Discussion may be streamlined (for example, it is not necessary to state that ATP is called the energy currency of the cells, lines 247-248).
8. Discussion (lines 248-259) should clarify that while glucose oxidation occurs to produce ATP, fatty acid oxidation is the main source of ATP in the heart

Reviewer 2 ·

Basic reporting

no comment

Experimental design

no comment

Validity of the findings

no comment

Additional comments

The manuscript by Tianliang Hou et al. describes an interesting phenomenon that provides mechanistic information for myocardial protective effects of Sevoflurane preconditioning following H9C2 cardiomyocyte hypoxia/reoxygenation model.
The authors demonstrate Sevoflurane preconditioning significantly increased the expression of HIF-1a, PDK-1, and Bcl-2 and reduced the protein expression of Bax, which markedly decreased the apoptosis ratio and Lactate dehydrogenase (LDH) level, increasing the cell viability, content of key enzymes of glycolysis, ATP concentration of glycolysis and stabilizing the mitochondrial membrane potential.
This study designed well, and the authors’ hypothesis is thoroughly tested. Their conclusions are interesting demonstrates that the cardioprotective effects of Sevoflurane preconditioning are potentially associated with the activation of the HIF-1a/PDK-1 signaling pathway.
My main concern is that thought the manuscript, SPC and HIF-1a inhibitor YC-1 were used as intervention methods for grouping study. Western Blot protein detection technique was used to quantify the content of target protein HIF-1a and PDK-1. The results showed that the contents of HIF-1a and PDK-1 in the SPC group increased significantly, while the contents of HIF-1a and PDK-1 decreased significantly in the S+Y group. The principal claim of the result is that the cardioprotective effects of Sevoflurane preconditioning are associated with the activation of the HIF-1a/PDK-1 signaling pathway.
I suggest the authors’ study confirmed that sevoflurane preconditioning can effectively attenuate the
Hypoxia/Reoxygenation (H/R) injury in H9c2 cells.

---

## Round 0.2 · Minor Revisions

Dear Dr. Hou,

I have reviewed the revised manuscript and rebuttal letter and believe it will be of interest to the readers of PeerJ. However a number of modifications to the figures and text need to be made before we can accept the manuscript for publication.

There appears to have been an issue with uploading the figures in the revised document (parts of Fig2 are missing, figures do not correspond with figure legends). Please re-upload these files and carefully proof the assembled PDF.

Several changes requested by the reviewers that are answered in the rebuttal letter did not make it into the revised manuscript. These include the labeling of Figure 1 to list T1/T2 and indicate the time frame for each experimental condition, and the updated labeling for the annexin studies in Fig2.

Additionally, please modify line 255 to indicate that most of the ATP in the heart comes from oxidative phosphorylation as requested in point 8 by reviewer 1.

Please review the labeling of the JC-1 study in Figure 4 (red and green appear to be swapped). Additionally, please indicate how many cells were scored for these studies in your methods section to enable rigor of study to be evaluated.

Please name the post-test associated with the ANOVAs in your statistics section.

The revised manuscript would benefit from review from someone with an eye for proper grammar and English usage as noted previously. Specifically, revise content to improve readability in lines 264-269. Carefully review the work for typographical errors. By convention, leave a single space between the word and parenthesis. Example: Hypoxia/Reoxygenation (H/R) not Hypoxia/Reoxygenation(H/R). This is an error throughout the work. Review for improper use of capitalization (ex: lines 64, 287, 295). Please revise font size of author names to match surrounding text (lines 308-315).

I look forward to receiving your revised manuscript.

---

## Round 0.3 · accepted · Accept

The revised manuscript sufficiently addresses the critiques of the two expert Reviewers. Congratulations!